# Unveiling the psychological traits of multi-marathoners: Insights from TIPI personality trait analysis

**Leo Lundy**[1]*, **Richard B. Reilly**[1,2,3], **Neil Fleming**[1,4], **Dominika Wilczyńska**[5]

1 Trinity Department of Mechanical, Manufacturing and Biomedical Engineering, Trinity College, The University of Dublin, Ireland, 2 School of Medicine, Trinity College, The University of Dublin, Ireland, 3 School of Engineering, Trinity College, The University of Dublin, Ireland, 4 Human Performance Laboratory, Department of Anatomy, School of Medicine, The University of Dublin, Ireland, 5 Faculty of Social and Humanities, University WSB Merito, Gdańsk, Poland

* lundyl@tcd.ie

## Abstract

### Objectives

Multi-marathoners, athletes dedicated to completing 100+ marathons, represent a unique endurance sport subculture. This study examines their psychological traits using the Ten Item Personality Inventory (TIPI) and Latent Class Analysis (LCA) to identify personality-based profiles and subgroup differences.

### Methods

An online cross-sectional survey of 593 multi-marathoners (56% men, 44% women, mean age = 53.87, SD = 9.91, countries =22) collected TIPI data. Reliability was assessed using Cronbach's Alpha and Guttman's Lambda 6. Statistical analyses included Mann-Whitney U tests, ANOVA Aligned Rank Transform (ART), Wilcoxon post-hoc tests, and Spearman's correlations to examine personality differences across gender, age and health variables. LCA identified distinct personality subgroups. Normative TIPI data served as a comparison benchmark.

### Results

Multi-marathoners exhibited higher conscientiousness ($F(1,591) = 2.42$, $p < 0.001$) but lower emotional stability ($F(1,591) = 5.525$, $p < 0.001$) than the general population, suggesting strong goal-directed behaviour but challenges in stress management. Women showed higher agreeableness ($W = 50809$, $p < 0.00091$), while age-related differences were not statistically significant. LCA revealed four personality-based subgroups, including those with high resilience and others with health vulnerabilities.

### Conclusion

Multi-marathoners display distinct psychological traits, particularly high conscientiousness and lower emotional stability. These findings highlight the need for tailored psychological

**Data availability statement:** All relevant data are within the manuscript and its Supporting Information files.

**Funding:** The author(s) received no specific funding for this work.

**Competing interests:** The authors have declared that no competing interests exist.

**Abbreviation:** FREC, Faculty of Health Sciences Research Ethics Committee; TIPI, Ten Item Personality Inventory.

interventions to support multi-marathon athletes' participation and well-being. Future research should explore longitudinal patterns and explore the efficacy of psychological interventions to enhance participation and well-being.

## Introduction

Multi-marathoning, the pursuit of completing multiple marathon events, represents a unique subculture within endurance sports. The defining achievement for many multi-marathoners is reaching the milestone of completing 100 marathons, a feat that requires extraordinary dedication, perseverance, and resilience [1]. While a wider-ranging observational study has documented the participatory nature of multi-marathoning [2], the psychological dimensions of this endurance sport are less understood.

This study is framed as an academic exploration of multi-marathoning, documenting the sport while integrating established theoretical frameworks, including the Big Five Personality Model, Goal Setting Theory (GST), Self-Determination Theory (SDT), and the Health Belief Model (HBM). By adopting this framing, the research examines the psychological mechanisms driving multi-marathoners' participation, health behaviours, and resilience. Through the identification of personality-driven subgroups, the study lays the groundwork for tailored intervention strategies aimed at enhancing well-being, sustaining participation, and addressing the distinctive challenges faced by this unique athletic community.

Endurance athletes often experience significant psychological challenges, including heightened stress, anxiety, and depressive symptoms, which can impact both performance and sustained participation [3–5]. Examining the personality traits associated with multi-marathoners provides insight into factors that may either hinder or enhance their resilience and motivation.

Personality refers to an individual's consistent patterns of feelings, thoughts, and behaviours. It is shaped by both genetic and environmental influences, integrating cognitive, emotional, and behavioural processes that contribute to individual differences. Trait theory, developed by Allport, Eysenck, and Cattell, provides a foundation for understanding personality structure [6–8].

This study integrates four theoretical frameworks to examine the psychological mechanisms underpinning multi-marathoning: the Big Five Personality Model, Goal Setting Theory (GST), Self-Determination Theory (SDT), and the Health Belief Model (HBM).

Research in sports psychology indicates that specific personality profiles are more conducive to endurance sports success, particularly those high in conscientiousness and low in emotional stability [9,10]. This aligns with trait theory which categorises personality into five core dimensions: openness, conscientiousness, extraversion, agreeableness, and emotional stability (neuroticism) [10,11]. These traits influence goal setting, motivation, and resilience in endurance athletes, with conscientiousness being particularly relevant to sustained training and achievement [12].

Goal Setting Theory (GST) proposes that specific, challenging goals enhance performance, provided they are accompanied by feedback and commitment [13]. Multi-marathoners exemplify this principle through their structured pursuit of completing 100 or more marathons, highlighting the role of personality traits in maintaining persistence and long-term motivation [14].

Self-Determination Theory (SDT) differentiates between intrinsic and extrinsic motivation, both of which are central to endurance participation. Intrinsically motivated athletes engage in multi-marathoning for personal fulfilment and challenge, while extrinsic factors, such as social recognition and community belonging, further reinforce participation. Personality

traits, particularly conscientiousness and extraversion, shape how these motivational forces drive sustained engagement [15].

The Health Belief Model (HBM) explains how individuals assess health risks, benefits, and barriers when making behavioural decisions [16,17]. In the context of multi-marathoning, this model provides insight into how personality traits influence health-related behaviours, injury prevention, and recovery strategies. For instance, conscientious individuals, known for their disciplined and structured approach, may be more likely to adhere to injury prevention strategies and structured training plans, whereas those with lower emotional stability may perceive greater barriers to participation due to stress or injury concerns [18].

By combining these frameworks, this study examines how personality traits interact with goal setting, motivation, and health perceptions, offering insights into the psychological resilience and behaviours of multi-marathoners.

Research on personality in endurance athletes has largely centred on marathoners, with little attention given to multi-marathoners. A systematic review conducted as part of this study of PubMed and Web of Science identified over 12,000 articles on marathoning, yet none explicitly examined multi-marathoning, highlighting its rarity in the literature [18]. Studies suggest that marathoners tend to exhibit psychological traits associated with mental resilience, including lower levels of negative emotions such as tension and fatigue, alongside heightened vigour and adaptability [19].

Marathoners are often characterised by strong mental health, self-discipline, and a capacity for sustained effort, with attributes such as intelligence, imagination, and self-sufficiency playing a role in endurance performance [12]. Psychological differences related to sex and age have also been noted, with findings indicating that endurance athletes may exhibit distinct trait variations over time [20].

Goal setting plays a crucial role in endurance sports, with effective self-regulation and structured goals linked to enhanced performance and resilience [21]. Athletes who demonstrate high conscientiousness and emotional stability are more likely to sustain long-term training and competition, while traits such as openness and extraversion may influence motivation and engagement [22,23]. Personality traits continue to be explored as key factors in understanding endurance success, underscoring the importance of psychological adaptability in high-performance settings.

This study employs validated psychological tools, such as the Ten Item Personality Inventory (TIPI), to assess personality traits. Supplementary data from a previous observational study on multi-marathoners enriches the analysis, offering insights into key aspects such as motivation, injuries, recoveries, diet, and chronic health conditions [2]. Advanced statistical techniques, including LCA, were used to identify subgroups with shared personality traits, further enhancing the understanding of risks and strengths within this population.

While longer forms, such as the 240-item NEO Personality Inventory or the 44-item Big Five Inventory [24,25], would enhance internal consistency, their length is impractical for online surveys. These instruments take 15 to 45 minutes to complete, which would likely lead to increased respondent fatigue and lower response rates [26], significantly affecting the statistical power of the study. In contrast, shorter forms like the Ten-Item Personality Inventory (TIPI) are ideal for large-scale internet-based studies, where brevity is critical.

Short-form measurement instruments have been validated extensively. For example, Gosling et al. (2003) validated the TIPI using a dataset of over 300,000 respondents [27]. Additionally, short forms such as the Mini-IPIP, BFI-S, and BFI-10 have been validated in large datasets such as the German Socio-Economic Panel (SOEP), a longitudinal study with over 20,000 participants [28,29]. These studies demonstrate that short forms can provide reliable assessments of the Big Five traits in large, diverse populations.

While there has been limited psychological analysis of multi-marathon runners, short-form Big Five personality instruments have been validated in similar endurance sports populations. Studies by Zeiger and Zeiger (2018) used short-form personality measures to profile mental toughness in endurance athletes, such as ultra-marathoners, who face comparable physical and psychological demands [30]. Similarly, Laborde et al. (2016) investigated personality traits in athletes using short form measures across both individual and team sports [31]. These validations support the use of brief personality inventories like TIPI for capturing relevant psychological traits in multi-marathoning.

This trade-off between brevity and internal consistency is well-documented in the literature [32,33]. Given the scope and scale of our study, we accepted this trade-off, and TIPI was the most practical choice.

By utilising TIPI [34], this study aims to describe specific personality traits and understand how they influence key psychological outcomes, such as goal-setting behaviours (such as achieving 100 marathon completions) and overall well-being (physical and mental health), by integrating the Big Five Personality Model with GST, SDT and the HBM, providing an analysis of the psychological factors that contribute to both the challenges and successes experienced by multi-marathoners. Additionally, the study explores how these personality traits may differ from those in the general population and how they might relate to sustained participation in multi-marathoning events.

Based on these theoretical frameworks and recent literature, This study hypothesises that multi-marathoners exhibit distinctive personality traits compared to the general population, particularly higher levels of conscientiousness and extraversion, alongside challenges in emotional stability and adaptability [35,20,36]. These traits are expected to drive their sustained participation and performance outcomes in the sport [20,37]. Through LCA, this study further hypothesises the emergence of meaningful subclasses within the multi-marathoner population, characterised by unique combinations of personality profiles, health outcomes, and motivational drivers, such as high openness linked to mental resilience and low emotional stability linked to health vulnerabilities.

It is further hypothesised that gender and age moderate the relationships between personality traits and health or motivational outcomes. Women are expected to demonstrate higher agreeableness and lower emotional stability than men, potentially influencing psychological resilience and subgroup membership [19,36]. Age is hypothesised to correlate with increased emotional stability and conscientiousness, shaping subclass distinctions in health and participation patterns [36,38].

Moreover, personality traits are expected to predict health-related behaviours and outcomes. For example, high conscientiousness and emotional stability are hypothesised to correlate with better injury recovery and chronic condition management, while low emotional stability may increase susceptibility to stress-related injuries [11,20]. Intrinsic motivation (e.g., personal achievement) is hypothesised to align with high openness and conscientiousness, while extrinsic motivation (e.g., social recognition) aligns with high extraversion and agreeableness [13,15,37].

Additionally, resilience and stress management capacity are hypothesised to differ across subclasses, with those scoring high in conscientiousness and emotional stability demonstrating greater coping ability and sustained participation [39].

By integrating these findings, the study deepens understanding of the psychological mechanisms underlying multi-marathoning, offering insights into tailored interventions for enhancing resilience, health outcomes, and sustained engagement in this unique endurance sport. The framing of this research extends beyond documenting the sport academically by

uncovering subgroup patterns and highlighting the clinical and practical implications of personality traits for multi-marathoners. These findings are intended to guide the development of interventions that enhance psychological resilience, support sustained participation, and address the unique challenges faced by this community. In doing so, this research contributes to the broader understanding of endurance sports and the distinctive traits that drive multi-marathoners to achieve extraordinary feats.

## Methods

### Study design

A cross-sectional survey, including the Ten Item Personality Inventory (TIPI) test, was conducted online via the Qualtrics platform [34,40], offering broad geographical reach and convenient participation. Responses were collected from multi-marathoners worldwide during the study period, from December 14, 2023, to March 31, 2024.

Inclusion criteria required respondents to be individuals who had either achieved the goal of completing 100 marathons or had set this as their running goal and were actively engaged in the sport. Exclusion criteria disqualified individuals who had minimal marathon experience and had completed fewer than two marathons.

Gatekeepers were employed to distribute the survey and were identified through an analysis of the structure of multi-marathoning at national and international levels. The study team approached all national and international multi-marathon clubs and event companies based in the UK and Ireland. All major global multi-marathon clubs and UK/Ireland-based event companies agreed to support the survey distribution through their social media channels, email groups, and newsletters, which were generally accessible to anyone interested in the sport.

### Handling of missing data

The survey platform enforced data completion, ensuring that all submitted responses were fully recorded. Only 100% completed surveys were used for analysis. Consequently, there were no cases of missing data, and all analyses were conducted on a complete dataset.

### Personality traits

The Ten-Item Personality Inventory (TIPI) used in this study measured five personality traits (Extraversion, Agreeableness, Conscientiousness, Emotional Stability, and Openness) on a 7-point Likert scale (1 = Strongly Disagree to 7 = Strongly Agree). Each trait was assessed using two items: one positively worded and one negatively worded. Negative items were reverse scored to ensure consistency in the directionality of responses.

The two item scores for each trait were aggregated by averaging their values. This aggregation produced a composite score for each trait, reflecting a spectrum of responses based on the Likert scale. These scores formed the basis for subsequent analysis, including LCA. TIPI is a validated tool widely used in psychological research for reliability [34].

Fig 1. shows the TIPI items gathered in this study.

### Procedure

Respondents accessed the online survey via a secure link distributed through social media channels and email lists of multi-marathon clubs and event organisations. Participation was voluntary and anonymous, with informed consent obtained from all respondents via the survey platform before their involvement, ensuring ethical standards were upheld.

Here are a number of personality traits that may or may not apply to you. You should rate the extent to which the pair of traits apply to you, even if one characteristic applies more strongly that the other.

I see myself as:

| | Disagree Strongly (1) | Disagree Moderately (2) | Disagree a Little (3) | Neither Agree nor disagree (4) | Agree a little (5) | Agree moderately (6) | Agree strongly (7) |
|---|---|---|---|---|---|---|---|
| Extraverted, enthusiastic. | O | O | O | O | O | O | O |
| Critical, quarrelsome. | O | O | O | O | O | O | O |
| Dependable, self-disciplined. | O | O | O | O | O | O | O |
| Anxious, easily upset. | O | O | O | O | O | O | O |
| Open to new experiences, complex. | O | O | O | O | O | O | O |
| Reserved, quiet. | O | O | O | O | O | O | O |
| Sympathetic, warm. | O | O | O | O | O | O | O |
| Disorganized, careless. | O | O | O | O | O | O | O |
| Calm, emotionally stable. | O | O | O | O | O | O | O |
| Conventional, uncreative. | O | O | O | O | O | O | O |

**Fig 1. TIPI test.**

## Statistical analysis

Data were analysed using R, a statistical software package [41]. Descriptive statistics were computed to summarise the demographic characteristics of the sample and the distribution of TIPI scores [34].

Normative data from the original TIPI study was used as a benchmark for comparison by age group and gender [27]. This provided a large relevant normative dataset (>300,000) for the Big Five personality dimensions in the general population grouped by gender and age group.

Due to the mixed nature of the data, assumptions of normality and homoscedasticity were not applied, and statistical analysis tests were chosen that were appropriate to the mix of data under analysis.

Cronbach's Alpha and Guttman's Lambda 6 were calculated to assess the internal consistency and reliability of the survey. Mann-Whitney U tests were conducted on TIPI traits, which were ordinal in nature, to ascertain if there was statistical significance of any differences between survey and normative datasets [42]. Mann-Whitney U tests were performed across age groups by gender (men = 1, women = 2) between the survey dataset and the normative

dataset [12,42,43]. For all Mann-Whitney U tests, Bonferroni corrections were applied to account for multiple comparisons, ensuring that the reported significance levels remained robust. Permutation tests (10,000 iterations) were applied to derive empirical null distributions for non-parametric test statistics, addressing potential violations of normality and homoscedasticity. Permutation-based and conventional non-parametric p-values were both reported. ANOVA Aligned Rank Transform (ART) Tests were used to examine the effects of gender and age group on various personality traits [44]. Wilcoxon Rank-Sum Post-hoc tests identified specific pairwise comparisons with significant differences, when comparing different genders and age groups [45]. An effect size of 0.5 was consistently utilised across all analyses. This value represents a moderate effect size, as defined by Cohen's conventions, and was chosen to balance sensitivity with interpretability in the context of our study.

To enrich the analysis and enhance the robustness of the findings, data from a separate, previously published observational study were integrated [2]. This study was conducted in 2023, ensuring contemporary relevance to the multi-marathoner population. It surveyed 826 multi-marathoners across 40 countries, provided a detailed profile of the community, capturing demographic characteristics, dietary patterns, motivational factors, health conditions, and injury histories. The average marathon completion count was 146.54 (SD 201.83) per respondent, with participants comprising 60.69% men (average age = 51.6, SD 9.96 years) and 39.3% women (average age = 48.83, SD 9.15 years). This rich dataset complements the TIPI analysis by providing a broader context for understanding personality traits in relation to key behavioural and health variables.

This already published study employed a range of survey question formats tailored to capture various aspects of the multi-marathoner experience. These included multiple-choice questions that allowed participants to select multiple answers from predefined lists for data variables involving motivations, diet, injuries and recoveries, binary (yes/no) responses for categorical variables like health outcomes. Demographic variables included age, gender, and the number of marathon completions were self-reported by respondents and captured in fields that incorporated inherent number-checking to ensure accuracy and prevent invalid entries.

This structured approach ensured comprehensive data collection while maintaining consistency and clarity across diverse participant groups.

Key variables from the observational dataset included dietary patterns such as plant-based diets (vegan, vegetarian) and omnivorous diets (regular, paleo, Pesco-vegetarian), motivational factors including achievement, competition, lifestyle, and social enjoyment, and health variables encompassing chronic conditions like heart issues, high blood pressure, tendon, and bone problems. Injury and recovery data included joint and muscle injuries, severe conditions such as stress fractures and heat exhaustion, and minor issues like blisters and chafing, along with recovery methods such as self-care, medication, and alternative therapies.

In all cases demographic variables included age, gender, and the number of marathon completions were collected by the survey platform. These were mandatory and self-reported by respondents in fields that incorporated inherent number-checking to ensure accuracy and prevent invalid entries.

For this study, these data were aggregated into correlation groups and aligned with TIPI personality traits, stratified by gender and age to ensure consistency. Statistical analyses were conducted to explore associations between TIPI traits and multi-marathon-related variables, employing robust methods to ensure the validity and reliability of findings. Spearman's Rank Correlation was used to evaluate the relationships between TIPI traits and the additional variables, with confidence intervals calculated to provide precision and reliability for all correlation estimates. To address potential violations of normality and homoscedasticity,

permutation tests with 10,000 iterations were applied to generate empirical null distributions for test statistics. To account for multiple comparisons, p-values were adjusted using Bonferroni corrections where appropriate, with significant results reported at thresholds of significant results at p < 0.001 subject to statistical correction.

LCA was conducted to identify distinct subgroups within the sample, revealing unique personality and demographic patterns. These subgroups were further analysed to assess differences in personality traits and their associations with key variables, particularly those showing strong or very strong correlations.

Item-response probabilities (**Pr(x)**) were derived to represent the likelihood of specific responses for personality traits and health indicators across the identified latent classes. While the Likert scale for individual TIPI items ranged from 1 to 7, the reverse scoring and averaging process resulted in response probabilities that extended beyond this range.

In this analysis, **Pr(x)** represents the probability of a specific response category or aggregated score for a given item or variable within each latent class. For example, **Pr(5)** corresponds to the likelihood of selecting the fifth category on the response scale, or an aggregated value of 5 based on the scoring process. These probabilities provide insight into the defining characteristics of each latent class by quantifying response patterns for personality traits and health indicators.

The study was conducted in accordance with the Declaration of Helsinki and received ethical approval from Trinity College Dublin's Faculty of Health Sciences Research Ethics Committee (FREC) Number 231005.

## Results

### Participants

A total of 593 responses were received in the cross-sectional survey of this study for TIPI, 56% men (n = 331, average age = 53.87 years, SD = 9.91), 44% women (n = 261, average age = 54.06, SD = 10.56). One respondent identified as another gender and was not included in the gender-based analysis. Respondents represented 22 different countries across six continents, with the most frequent participation from Ireland, UK, Germany, Finland, Italy, and the US. These countries are among those with high multi-marathon participation according to the multi-marathoning world rankings [46]. The mean marathon completion count was 146.52 (SD:196.3), indicating that respondents were experienced multi-marathoners. A resolution measure of ± 1 year for age and ± 1 marathon for the number of marathons completed was applied, acknowledging minor variances in self-reported data while maintaining precision for analysis.

### Personality traits

TIPI scores were analysed and compared to normative data from the original TIPI study [27]. Fig 2 presents the results of this comparison versus TIPI norms. Statistically significant negative differences were observed in agreeableness and emotional stability across all age groups and genders. Higher conscientiousness was evident in women compared to TIPI norms.

Table 1 presents the means and standard deviations (M/SD) for each TIPI survey personality trait by age group and gender, along with the comparisons against the TIPI normative dataset.

Cronbach's alpha (0.42) and Guttman's Lambda 6 (0.40) indicated lower reliability compared to longer-form rating instruments [32–34,42]. However, this level of reliability is consistent with ultra-brief scales like the TIPI, which assesses five traits using only ten items.

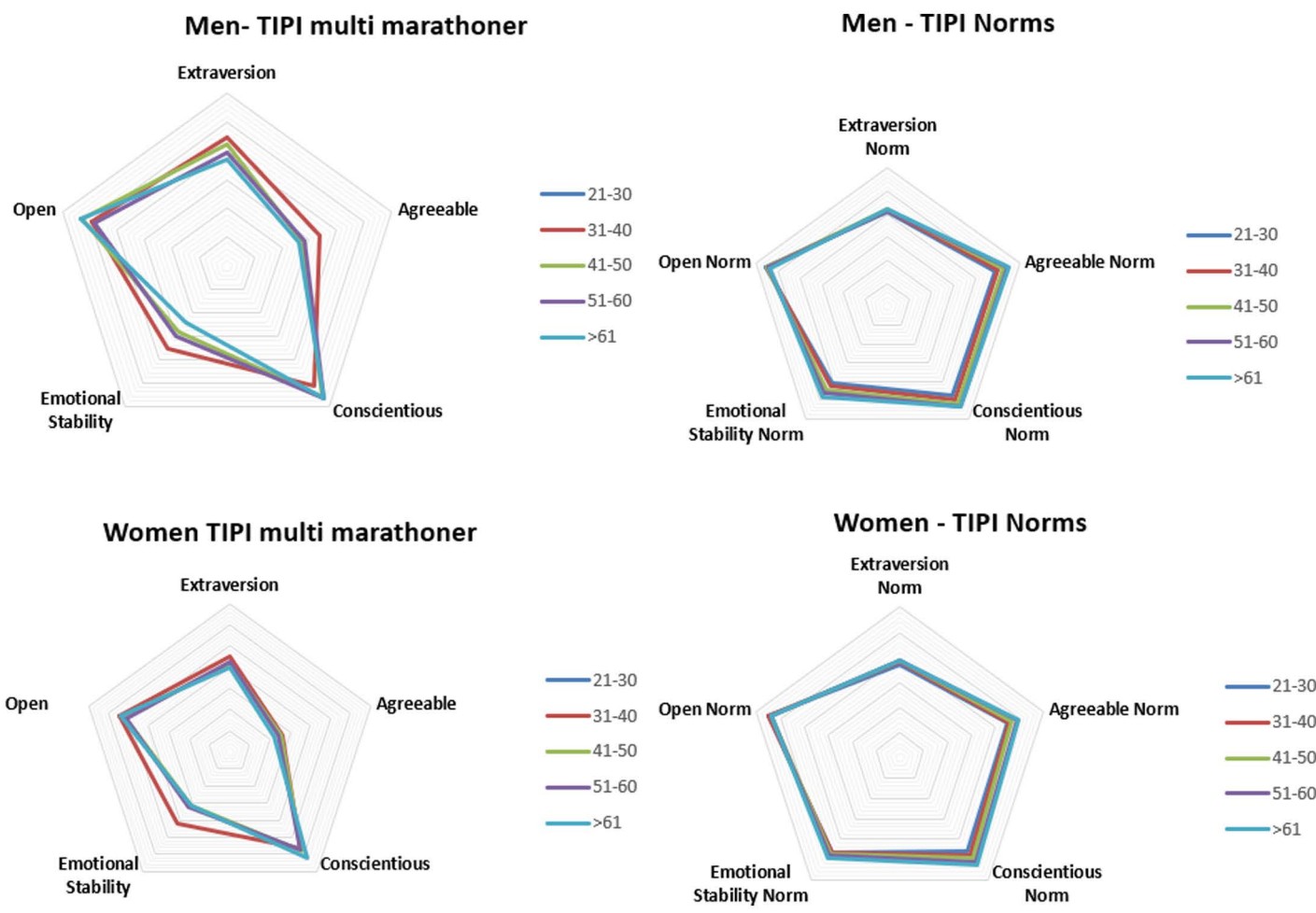

**Fig 2. Multi-marathon TIPI results versus TIPI norms.**

Mann-Whitney U tests identified statistically significant differences in agreeableness, conscientiousness, and emotional stability with α=0.05 and a Bonferroni correction applied for 50 comparisons (5 traits across 10 age groups). Results with p-values below 0.001 were considered significant and are denoted as < 0.001.

Table 2 presents the p-values and permutation-based results (10,000 permutations) from the Mann-Whitney U tests for all TIPI traits compared against the normative dataset.

## ANOVA aligned rank transform results

ANOVA Aligned Rank Transform (ART) tests were conducted to examine the effects of Gender and Age Groups on TIPI traits. Significant differences were observed in extraversion, agreeableness, emotional stability, and openness due to age group effects. In the case of conscientiousness, significant effects of both gender and age group were found. No significant interaction effects between gender and age group were observed for any traits. α= 0.05 with p-values below 0.001 were considered significant and are denoted as < 0.001.

Table 3 shows the results of these ANOVA ART tests.

Given the significant ANOVA ART results, Wilcoxon rank-sum post-hoc tests were applied for pairwise comparisons of personality traits across gender and age groups. A

**Table 1. Comparison of Survey TIPI results against TIPI Normative data.**

| Men | | | | | | |
|---|---|---|---|---|---|---|
| **Age Group** | | **21-30** | **31-40** | **41-50** | **51-60** | **61 +** |
| n | Survey | 4 | 27 | 83 | 132 | 85 |
| | Normative | 46530 | 15412 | 8823 | 4135 | 885 |
| Extraversion (M/SD) | Survey | (4.75/1.41) | (4.46/1.07) | (4.22/1.22) | (3.93/1.13) | (3.71/0.98) |
| | Normative | (4.07/1.61) | (4.17/1.64) | (4.2/1.64) | (4.18/1.6) | (4.21/1.62) |
| Agreeableness (M/SD) | Survey | (3.63/2.30) | (3.37/1.15) | (2.75/1.33) | (2.82/1.07) | (2.61/0.99) |
| | Normative | (4.88/1.19) | (5.04/1.19) | (5.28/1.17) | (5.43/1.14) | (5.5/1.15) |
| Conscientiousness (M/SD) | Survey | (6.13/0.88) | (5.11/1.10) | (5.63/1.15) | (5.67/0.88) | (5.69/0.90) |
| | Normative | (4.78/1.41) | (4.97/1.41) | (5.18/1.36) | (5.35/1.31) | (5.39/1.36) |
| Emotional Stability (M/SD) | Survey | (3.25/1.06) | (3.52/0.89) | (2.83/0.86) | (3.01/0.87) | (2.41/0.62) |
| | Normative | (4.09/1.45) | (4.25/1.45) | (4.49/1.45) | 4.66/1.44) | (4.84/1.4) |
| Openness (M/SD) | Survey | (5.38/0.88) | (4.95/1.07) | (5.28/1.12) | (4.85/1.18) | (5.34/1.06) |
| | Normative | (5.55/1.12) | (5.49/1.18) | (5.46/1.2) | (5.42/1.25) | (5.39/1.27) |
| Women | | | | | | |
| **Age Group** | | **21-30** | **31-40** | **41-50** | **51-60** | **61 +** |
| n | Survey | 3 | 19 | 83 | 88 | 68 |
| | Normative | 40737 | 14752 | 7668 | 3532 | 905 |
| Extraversion (M/SD) | Survey | (5.67/0.47) | (4.53/1.12) | (4.16/1.09) | (4.27/1.03) | (3.96/0.97) |
| | Normative | (3.73/1.54) | (3.81/1.54) | (3.85/1.54) | (3.87/1.54) | (3.85/1.49) |
| Agreeableness (M/SD) | Survey | (3.83/2.12) | (2.61/1.45) | (2.52/1.13) | (2.44/1.16) | (2.24/0.99) |
| | Normative | (4.5/1.2) | (4.55/1.21) | (4.7/1.18) | (4.89/1.18) | (4.95/1.17) |
| Conscientiousness (M/SD) | Survey | (5.67/0) | (5.63/0.67) | (5.83/0.93) | (5.72/1.05) | (6.19/0.67) |
| | Normative | (4.57/1.39) | (4.77/1.35) | (4.96/1.35) | (5.11/1.31) | (5.26/1.3) |
| Emotional Stability (M/SD) | Survey | (3.33/0.47) | (4.16/0.75) | (3.13/0.96) | (3.26/1.1) | (3.13/1.07) |
| | Normative | (4.64/1.46) | (4.63/1.42) | (4.72/1.39) | (4.8/1.38) | (4.92/1.34) |
| Openness (M/SD) | Survey | (5/0.94) | (5.5/1.38) | (5.23/1.04) | (5.15/1.22) | (5.43/0.94) |
| | Normative | (5.49/1.13) | (5.49/1.12) | (5.41/1.17) | (5.39/1.2) | (5.27/1.26) |

Note: M = Mean, SD = Standard Deviation

**Table 2. P-value/Permutation results for Mann Whitney U tests by age group and gender (men = 1, women = 2).**

| Age | Gender | Extraversion p-val/perm | Agreeableness p-val/perm | Conscientiousness p-val/perm | Emotional Stability p-val/perm | Openneness p-val/perm |
|---|---|---|---|---|---|---|
| 21-30 | 1 | 0.627/0.517 | 0.116/0.115 | 0.028/0.030 | 1/0.675 | 0.844/0.692 |
| 31-40 | 1 | 0.124/0.076 | <0.001/<0.001 | 0.588/0.831 | 0.056/0.039 | 0.365/0.11 |
| 41-50 | 1 | 0.924/0.900 | <0.001/<0.001 | <0.001/<0.001 | <0.001/<0.001 | 0.688/0.382 |
| 51-60 | 1 | 0.275/0.228 | <0.001/<0.001 | 0.008/0.020 | <0.001/<0.001 | <0.001/<0.001 |
| 61 + | 1 | 0.053/0.064 | <0.001/<0.001 | 0.239/0.373 | <0.001/<0.001 | 0.27/0.526 |
| 21-30 | 2 | 0.693/0.502 | <0.001/<0.001 | <0.001/<0.001 | 0.708/0.402 | 1/1 |
| 31-40 | 2 | 0.198/0.135 | <0.001/<0.001 | <0.001/<0.001 | 0.674/0.561 | 0.454/0.607 |
| 41-50 | 2 | 0.039/0.037 | <0.001/<0.001 | <0.001/<0.001 | <0.001/<0.001 | 0.976/0.848 |
| 51-60 | 2 | 0.096/0.101 | <0.001/<0.001 | 1/0.9 | <0.001/<0.001 | 0.402/0.26 |
| 61 + | 2 | 0.569/0.296 | <0.001/<0.001 | <0.001/0.033 | <0.001/<0.001 | 0.229/0.42 |

Note: p-val = p-value, perm = permutation p-value

Bonferroni correction was applied for 55 tests (5 traits across 10 age groups plus gender), with p-values below 0.001 considered highly significant.

Table 4 presents Wilcoxon rank-sum post-hoc test results.

## TIPI traits and key factors influencing multi-marathon participation

Spearman's correlation analysis was conducted to examine associations between TIPI traits and variables from a previously published observational study (as outlined in the Methods section), which surveyed 826 multi-marathoners across 40 countries.

Spearman's ρ was used to categorise correlations as: 0-0.19 (Very Weak), 0.2-0.39 (Weak), 0.4-0.59 (Moderate), 0.6-0.79 (Strong), and 0.8-1 (Very Strong). Seven instances of strong or very strong correlations were identified, as summarised in Table 5.

**Table 3. ANOVA ART results examining the effects of Gender and Age Group on Traits.**

| Trait | Gender.F | Gender.p | Age.F | Age.p | Interaction.F | Interaction.p | Effect.Type |
|---|---|---|---|---|---|---|---|
| Extraversion | 0.188 | 0.665 | 2.749 | 0.028 | 0.548 | 0.701 | Age Group |
| Agreeableness | 3.307 | 0.069 | 4.978 | <0.001 | 0.636 | 0.637 | Age Group |
| Conscientiousness | 4.593 | 0.033 | 2.421 | 0.047 | 1.506 | 0.199 | Age Group Gender |
| Emotional Stability | 2.072 | 0.151 | 5.525 | <0.001 | 0.774 | 0.542 | Age Group |
| Openness | 1.219 | 0.270 | 2.539 | 0.039 | 1.450 | 0.216 | Age Group |

Note: F = F-statistic, p = p-value, Gender.F = F-statistic for gender effect, Gender.p = p-value for gender effect, Age.F = F-statistic for age group effect, Age.p = p-value for age group effect, Interaction.F = F-statistic for gender × age interaction, Interaction.p = p-value for gender × age interaction.

**Table 4. Wilcoxon rank-sum post-hoc tests by Gender (men = 1, women = 2) and Age-Group.**

| Trait | Effect Type | W statistic | p-value |
|---|---|---|---|
| Agreeableness | Gender: 1 vs 2 | 50809 | <0.001 |
| Agreeableness | Age Group: 61 and older vs 21-30 | 191 | 0.004 |
| Conscientiousness | Gender: 1 vs 2 | 36638.5 | 0.001 |
| Emotional Stability | Gender: 1 vs 2 | 36956.5 | 0.002 |

Note: W statistic = Wilcoxon rank-sum test statistic, p-value = significance level for the Wilcoxon test. Gender: 1 vs 2 refers to comparisons between men and women.

**Table 5. Strong significance levels (Spearman's ρ) of correlations of TIPI Traits.**

| TIPI Trait | Correlated variables | Spearman's ρ | Permutation | CI lwr | CI upr |
|---|---|---|---|---|---|
| Extraversion | Health: High Blood Pressure | -0.62 | <0.001 | 0.65 | 0.86 |
| Agreeableness | Health: Chronic illness | -0.67 | <0.001 | -0.6 | -0.16 |
| Emotional Stability | Health: High Blood Pressure | -0.82 | <0.001 | 0.4 | 0.74 |
| Emotional Stability | Health: Balance | 0.7 | <0.001 | 0.24 | 0.65 |
| Extraversion | Injury: Minor Issue | 0.64 | <0.001 | 0.24 | 0.7 |
| Extraversion | Injury: Running Injury | -0.69 | <0.001 | -0.92 | -0.76 |
| Emotional Stability | Injury: Running Injury | -0.78 | <0.001 | -0.98 | -0.93 |

Note: Spearman's ρ = Spearman's rank correlation coefficient, Permutation = p-value obtained from permutation testing (10,000 permutations), CI lwr = Lower bound of the Confidence Interval, CI upr = Upper bound of the Confidence Interval.

### Latent class analysis (LCA)

LCA identified four distinct classes based on personality traits and highly correlated variables. Model selection was guided by fit indices, with the 4-class model demonstrating the best balance between complexity and interpretability (AIC = 16523.52, BIC = 18040.21). This model effectively captured meaningful subgroups within the sample.

Fig 3 presents the proportions of participants in each latent class.

### Item-response probabilities by class

Item-response probabilities provide insights into the key characteristics of each latent class, delineating variations in personality traits and health-related indicators.

Table 6 summarises key personality traits and health indicators for each class.

## Discussion

This study provides insights into the psychological traits and health profiles of multi-marathoners, leveraging both aggregate results from the Ten Item Personality Inventory (TIPI) [15,27] and detailed subgroup patterns identified through LCA.

**Fig 3. Proportions of participants in each latent class based on the 4-class LCA model.** The largest group, Class 2, comprises 41.7% of the sample, followed by Class 4 (31.6%), Class 3 (14.5%), and Class 1 (12.2%).

**Table 6. Key characteristics of the latent classes, showing personality and health-related profiles.**

| Class | Proportion of Participants | Key Personality Traits | Key Health Indicators |
|---|---|---|---|
| Class 1 | 12.2% | Moderate emotional stability, low openness | High probability of poor blood pressure outcomes (Pr(5) = 0.282), moderate mental health (Pr(8) = 0.352) |
| Class 2 | 41.7% | Balanced profile across traits | Favourable bone-to-tendon ratio (Pr(2) = 0.534), moderate mental health (Pr(4) = 0.534) |
| Class 3 | 14.5% | High openness (Pr(10) = 0.186), strong mental health (Pr(6) = 0.965) | Lower physical health outcomes such as blood pressure (Pr(3) = 0.965) |
| Class 4 | 31.6% | Moderate openness, high conscientiousness | Balanced mental health and physical health indicators |

*Note: Pr(x) = Probability of a given outcome occurring for a specific variable in the latent class analysis model. Percentages indicate the proportion of participants in each latent class.*

This study hypothesised that multi-marathoners exhibit distinctive personality traits compared to the general population. The findings strongly support this, revealing significantly higher levels of conscientiousness and lower levels of emotional stability. These traits align with the goal-oriented and disciplined behaviours essential for sustaining participation in this demanding sport. Conversely, lower emotional stability underscores the psychological challenges associated with stress management and emotional regulation, which could increase psychological strain in response to the physical and mental demands of the sport. While extraversion was observed, its direct influence on participation and social engagement warrants further exploration. Openness did not emerge as a central trait in the overall sample. However, its prominence in Class 3 individuals suggests that exploratory tendencies may be more relevant for resilience-focused subgroups, highlighting the importance of subgroup analysis in revealing trait expressions.

The study proposed that distinct subclasses would emerge through LCA. The results confirm this, with four unique subclasses identified. These classes range from individuals with high conscientiousness and mental resilience to those with moderate emotional stability and health vulnerabilities. For example, Class 1, characterised by moderate emotional stability and lower openness, faces significant cardiovascular risks, emphasising the need for targeted health interventions. Class 2, the largest subgroup, displays balanced personality traits and moderate health outcomes, suggesting a need for general health optimisation strategies. Class 3 individuals, with high openness and robust mental health, exemplify resilience and adaptability, traits that align well with the exploratory nature of multi-marathoning. Finally, Class 4 demonstrates high conscientiousness and balanced health, highlighting the utility of structured, goal-driven behaviours in sustaining participation and performance.

These subgroup-level analyses underscore the diversity within the multi-marathoner population and the importance of tailoring interventions based on subclass-specific characteristics. The identification of latent classes adds depth to the findings, revealing how subgroup-specific patterns of personality and health influence participation behaviours. This aligns with recent research demonstrating that elite and competitive athletes exhibit distinct personality clusters, reinforcing the importance of tailored psychological interventions [9].

The study investigated whether gender and age moderate the relationships between personality traits and health or motivational outcomes. Gender differences revealed that women exhibited significantly higher levels of agreeableness compared to men, reflecting tendencies toward cooperative and empathetic behaviours. These differences may influence social interactions and community dynamics within the sport, with potential implications for group-based training and engagement strategies. While no significant differences in conscientiousness or emotional stability were observed between genders overall, the results suggest

that gender and age may interact in complex ways. Future research could explore whether age moderates these traits differently for men and women, contributing to differences in participation and group dynamics.

Age-related differences were also explored, though they were not statistically significant after applying corrections for multiple comparisons. Older participants exhibited increased emotional stability, which may contribute to better stress management. These findings suggest that while gender and age play roles in shaping participation and health outcomes, their effects are nuanced and warrant further investigation into interactions between these variables.

The findings validate the proposal that personality traits predict health-related behaviours and outcomes. High conscientiousness was associated with better adherence to training regimens and effective injury management strategies. Conversely, lower emotional stability correlated with increased injury susceptibility and psychological strain. Subclass-specific analyses reinforced these findings, with resilience-focused groups exhibiting superior recovery and maintenance of physical health. These results highlight the critical role of personality in shaping health behaviours, particularly in managing the physical demands of multi-marathoning.

This study also explored how intrinsic motivation aligns with personality traits such as high openness and conscientiousness, while extrinsic motivation aligns with traits such as high extraversion and agreeableness. Intrinsic motivations, such as personal achievement and goal fulfilment, were strongly linked to conscientiousness. Extraversion contributed to social engagement and extrinsic motivation but was less influential than expected. Agreeableness appeared to enhance social dynamics but did not emerge as a primary driver of participation. Motivational patterns also varied across subclasses, with Class 4's high conscientiousness aligning more with intrinsic goals, while Class 2's balanced traits correlated with broader social motivations. These results underscore the multifaceted drivers behind sustained participation in endurance sports and point to opportunities for further research on the interplay between motivational patterns and personality traits.

The results highlight that subclasses with high conscientiousness and emotional stability demonstrate greater resilience and stress management capacity. High-resilience subclasses exhibited superior coping strategies and sustained participation. Conversely, lower-resilience groups, characterised by low emotional stability, faced greater psychological challenges, emphasising the importance of tailored mental health support. These findings highlight the necessity of resilience-building interventions to address the unique needs of specific subclasses, such as stress management workshops for low-resilience groups or targeted motivational training for those struggling with goal adherence.

By integrating these theoretical models, the study provides a robust framework for understanding how personality traits interact with motivational and health-related factors. This comprehensive approach not only captures the psychological dimensions of multi-marathoning but also offers actionable insights for fostering sustained engagement and optimal performance.

## Practical implications

The results have practical applications for designing interventions to support the well-being and performance of multi-marathoners. Tailored health interventions, such as cardiovascular screenings for Class 1 or mental health resources for individuals with lower emotional stability, can address specific vulnerabilities. Structured training programmes and goal-setting workshops can leverage the strengths of high-conscientiousness groups, such as Class 4, to optimise their performance. Goal setting plays a crucial role in endurance sports, with effective self-regulation and structured goals linked to enhanced performance and resilience [47].

Research suggests that athletes high in conscientiousness and emotional stability are more likely to sustain long-term training and competition, while openness and extraversion influence motivation and engagement [48].

Community engagement strategies that foster social connections and peer recognition, particularly for balanced-profile groups like Class 2, can enhance motivation and sustained participation. Additionally, resilience-building programmes targeting stress management and emotional regulation can mitigate the challenges associated with lower emotional stability, fostering a healthier and more balanced approach to the sport.

The HBM framework further supports the development of health-focused interventions by identifying key perceptions that drive health behaviours. By addressing perceived barriers and enhancing the perceived benefits of proactive health management, interventions can empower multi-marathoners to adopt healthier practices and sustain their participation over time.

Until now, the absence of peer-reviewed research has resulted in a lack of understanding of nature of the sport.

Multi-marathon participants and those involved in multi-marathon governance, in the provision of multi-marathon events, or health professionals, may utilise the recommendations given in this study to better plan their contribution to the sport and its overall safety, policies and organisation.

## Future research directions

Future research should explore how personality traits evolve with continued participation in multi-marathoning, using longitudinal studies to examine trait stability and dynamic changes over time. Such research could clarify whether resilience for example is a stable personality trait or a context-dependent capacity shaped by external challenges and support systems. Investigating the impact of specific interventions, such as tailored training programs, stress management techniques, and mental health support, could provide evidence-based strategies to enhance resilience and mitigate challenges associated with low emotional stability.

Expanding the diversity of study samples to include broader cultural, socioeconomic, and geographic contexts is essential for improving the generalisability of findings. This would also provide a more nuanced understanding of how external factors, such as access to resources or cultural attitudes, influence participation and personality expression in endurance sports.

Future studies should also validate and refine personality assessment tools, comparing brief measures like the Ten Item Personality Inventory (TIPI) to longer instruments, such as the NEO-PI-R. This would ensure the accuracy and reliability of personality measurement in multi-marathoners. Additionally, tracking subgroup dynamics identified through Latent Class Analysis (LCA) could reveal how health outcomes, motivation, and personality profiles evolve over time, informing tailored intervention strategies for distinct athlete profiles.

By addressing these areas, future research can advance understanding of the psychological and contextual factors that underpin multi-marathoning, providing a foundation for more effective support and interventions in this unique endurance sport.

## Limitations

While the study aimed to maintain respondent anonymity and reduce potential biases, several limitations should be noted:

**Selection Bias** The survey distribution through gatekeepers in closed social media groups might introduce selection bias, affecting the generalisability of the results.

**Response Bias** Using anonymous self-reported data may lead to response bias, where participants provide socially acceptable or inaccurate answers.

**Unpublished data** Use of unpublished TIPI norms provided by Gosling et al. for comparison purposes [27]. While these norms are widely referenced in personality research, their unpublished status means that they may not have been subjected to the same rigorous peer review as published data. This could introduce some uncertainty regarding the accuracy or representativeness of the normative data used as a benchmark in this study.

**Lack of Insights from Dropouts** The survey did not include insights from individuals who stopped pursuing multi-marathoning, limiting understanding of factors influencing participation and retention.

**Brief scale** TIPI is known as an ultra-brief scale with a trade-off between brevity and internal consistency. Use of ultra-brief scales does not provide the same level of depth as longer inventories and could introduce lower reliability than if a long-form personality test was used.

**Language** A limitation of the study is the use of multiple validated translations in the survey. 78% of responses were completed in English (TIPI's base language). However, minor differences in interpretation across languages may introduce some variability. Predominant use of English, use of validated translations and the straightforward nature and brevity of TIPI mitigate this concern [49–52]

**Measurement Instrument** the study focused exclusively on the Big Five personality traits, as measured by the Ten Item Personality Inventory (TIPI). While TIPI is a validated and widely used tool for assessing personality, its brevity may limit the depth and granularity of the data compared to more comprehensive measures, such as the NEO-PI-R or BFI-44.

**Demographic variables** were limited to age, gender, and the number of marathon completions. The study did not control for potentially confounding variables, such as socio-economic status, cultural background, or access to resources (e.g., coaching or training facilities), which could influence participation, personality expression, and health outcomes. These unmeasured variables may have introduced bias or limited the generalisability of the findings.

Despite these limitations, the study offers important insights into different aspects of multi-marathoning, setting the stage for future research, which should consider longitudinal designs and objective measures of psychological health to build on the findings of this study.

## Conclusion

The findings of this study provide compelling evidence that multi-marathoners exhibit distinctive personality profiles, characterised by high conscientiousness and lower emotional stability. These results strongly support the assertion that self-discipline is a key psychological factor enabling sustained multi-marathon participation, while lower emotional stability suggests potential vulnerabilities in stress management. The absence of significant differences in openness and the interaction between age and personality traits challenges prior assumptions and underscores the complex and multifaceted nature of personality influences in multi-marathoning.

The integration of Big Five Personality Theory, Goal-Setting Theory (GST), Self-Determination Theory (SDT), and the Health Belief Model (HBM) further reinforces the reliability of these findings by providing a robust theoretical foundation for understanding the psychological mechanisms at play. These insights offer strong support for the development of targeted interventions, such as structured training programmes, personalised goal-setting strategies, and stress management techniques that can enhance both the physical and psychological resilience of multi-marathoners.

By integrating these findings, the study deepens our understanding of the psychological mechanisms underlying multi-marathoning, offering insights into tailored interventions for enhancing resilience, health outcomes, and sustained engagement. Recent research

has reinforced the significance of personality traits in competitive sports performance, highlighting the role of conscientiousness, extraversion, and neuroticism in endurance success [5].

Furthermore, the latent class analysis (LCA) robustly identifies distinct subgroups within the multi-marathoner population, demonstrating clear variations in personality and health profiles. These findings strongly advocate for the necessity of tailored support strategies that recognise individual strengths while addressing specific vulnerabilities.

The study's methodology and statistical rigour further strengthen confidence in these conclusions. However, future research should critically examine longitudinal changes in personality traits, evaluate the effectiveness of psychological interventions, and refine the validation of short-form personality measurement instruments for this population. Expanding the demographic diversity of future samples will be essential to confirming the generalisability of these findings across the global multi-marathon community.

In conclusion, this study provides strong empirical support for the role of conscientiousness and emotional stability in shaping multi-marathon participation and performance. The application of theoretical frameworks and advanced analytical techniques reinforces the validity and impact of these findings. By advancing the psychological understanding of endurance athletes, this research lays the groundwork for evidence-based interventions that foster resilience, sustained engagement, and peak performance in multi-marathoners.

## Acknowledgments

The authors of this study would like to offer thanks to all study participants for their contribution to this research.

## Author contributions

**Conceptualization:** Leo Lundy, Richard B Reilly, Neil Fleming, Dominika Wilczyńska.

**Data curation:** Leo Lundy.

**Formal analysis:** Leo Lundy.

**Investigation:** Leo Lundy.

**Methodology:** Leo Lundy.

**Project administration:** Leo Lundy.

**Supervision:** Richard B Reilly, Neil Fleming.

**Visualization:** Leo Lundy.

**Writing – original draft:** Leo Lundy.

**Writing – review & editing:** Richard B Reilly, Neil Fleming, Dominika Wilczyńska.

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
