## [Decision Letter · Decision Letter 0]

18 Oct 2024

PONE-D-24-41718Unveiling the Psychological Traits of Multi-Marathoners: Insights from TIPI Personality Trait AnalysisPLOS ONE

Dear Dr. Lundy,

Thank you for submitting your manuscript to PLOS ONE. After careful consideration, we feel that it has merit but does not fully meet PLOS ONE’s publication criteria as it currently stands. Therefore, we invite you to submit a revised version of the manuscript that addresses the points raised during the review process.

Both reviewers recognized some positive aspects of your manuscript. However, in addition to some lacking references and problems with the visibility of the figures, there are two major concerns that your revision needs to address in order for the manuscript to be accepted:1. The lack of theoretical foundation - the theoretical framework needs to be more developed and include some potentially relevant theories such as Goal Setting theory2. The issue related with the measurement of Big Five Personality traits - that is using its short form that has not been validated before. This could be tackled by using the longer form on a smaller target population, or by validating the short form on any population. I recognize that doing that may require more time, but if possible (and you can ask for the extension of time needed for the revision), I would recommend you to do that. Alternatively, you need to provide a very strong rebuttal regarding this issue.3. You also need to address all minor suggestions of the reviewer.

We look forward to receiving your revised manuscript.

Kind regards,

Srebrenka Letina, Ph.D.

Academic Editor

PLOS ONE

3. Please upload a new copy of Figures 1 and 2 as the detail is not clear. Please follow the link for more information: https://blogs.plos.org/plos/2019/06/looking-good-tips-for-creating-your-plos-figures-graphics/" https://blogs.plos.org/plos/2019/06/looking-good-tips-for-creating-your-plos-figures-graphics/

Reviewers' comments:

Reviewer's Responses to Questions

**Comments to the Author**

1. Is the manuscript technically sound, and do the data support the conclusions?

Reviewer #1: Partly

Reviewer #2: Yes

2. Has the statistical analysis been performed appropriately and rigorously? 

Reviewer #1: Yes

Reviewer #2: Yes

3. Have the authors made all data underlying the findings in their manuscript fully available?

Reviewer #1: Yes

Reviewer #2: Yes

4. Is the manuscript presented in an intelligible fashion and written in standard English?

Reviewer #1: Yes

Reviewer #2: Yes

5. Review Comments to the Author

Reviewer #1: General comments:

The study presents a robust justification and a well-structured background. The objectives are clearly articulated and the design of the study is commendable. The methodology is thoroughly detailed, and the results derived from it are grounded in reliable instruments and rigorous statistical procedures.

Introduction:

The introduction is structured, delineating the existing knowledge gap that the current study aims to address. This articulation serves to elucidate the study's purpose in a coherent and scholarly manner.

Authors may benefit from further discussion regarding the application of both the Big Five Personality Model and Goal Setting Theory. Recent advancements in the understanding of human behavior indicate that various other theories and constructs can also elucidate behaviors related to physical activity. It remains ambiguous to me what substantial value is derived from involving 593 participants in a questionnaire grounded solely in these theories.

Additionally, there are several paragraphs that lack references. It is important to consider the necessity of citations throughout the background section.

Methods:

Methods are clear and allow the study to be repeated.

Concerns arise regarding the utilization of a questionnaire lacking prior validation for the specific population and language. While the presentation of the internal consistency and reliability of the survey is an important step, it concurrently underscores the shortcomings of the employed measures.

Results:

Despite the unsatisfactory methodology employed in obtaining the results, the findings successfully fulfill the objectives of the study.

Reviewer #2: This article presents solid content, is well-conceived, and properly structured and executed.

It offers highly relevant information about the personality traits of multi-marathon runners.

The data collection is extensive, with 593 responses, considering that this is a limited population, such as that of multi-marathon runners.

The statistical analyses are appropriate and very well conducted. They are carried out with an excellent level of detail, including the Bonferroni correction, which is not always performed.

Some results may be unexpected, such as in the dimension of openness to experience. However, for the rest of the results, the expected outcomes are essentially met.

In any case, the work provides an interesting perspective on the psychological characterization of multi-marathon runners. This is of great interest for practical intervention in counseling and psychological coaching for these athletes.

The main challenge of the work, as I understand it, lies in the use of the TIPI instrument to evaluate the Big Five personality dimensions. Although it is a sufficiently standardized and widely accepted tool by the scientific community in personality psychology.

In conclusion, this is a well-conceived and properly executed scientific study, offering relevant contributions to both the theory and practice of sports psychology.

6. PLOS authors have the option to publish the peer review history of their article (what does this mean? ). If published, this will include your full peer review and any attached files.

**Do you want your identity to be public for this peer review?** For information about this choice, including consent withdrawal, please see our Privacy Policy .

Reviewer #1: **Yes: ** Hugo Vieira Pereira

Reviewer #2: **Yes: ** Vanesa García-Peñas

---

## [Author Response · Author response to Decision Letter 1]

30 Oct 2024

A full rebuttal document has been uploaded and labeled "Response to Reviewers". This is very detailed and addresses each and every comment of the editor and each of the reviewers.

Also all data is available in the manuscript or uploaded in a supporting file (Excel spreadsheet) as part of this submission. This spreasdsheet has column labels that are in context with the analysis carried out.

---

## [Decision Letter · Decision Letter 1]

14 Nov 2024

PONE-D-24-41718R1Unveiling the Psychological Traits of Multi-Marathoners: Insights from TIPI Personality Trait Analysis

PLOS ONE

Dear Dr. Lundy,

Thank you for submitting your manuscript to PLOS ONE. After careful consideration, we feel that it has merit but does not fully meet PLOS ONE’s publication criteria as it currently stands. Therefore, we invite you to submit a revised version of the manuscript that addresses the points raised during the review process.

Two expert reviewers and I have read your paper, and we recognize that the manuscript addresses an interesting research question and that your revision has responded to the specific concerns raised. However, one of the reviewers has expressed concerns about the general scientific value and quality of the paper, which, in my opinion, also undermines the manuscript's value in its current form. Unfortunately, the reviewer did not provide specific suggestions for improving the paper, so I would like to suggest some avenues you might pursue to add the necessary value:

**Introduction and Literature Review:** Provide a more detailed review of the previous literature on athletes and Big Five personality traits, with a focus on how and why the personality profile of multi-marathoners may theoretically differ, and summarize what empirical studies have shown so far. A systematic analysis of previous research findings on this topic would greatly enhance the quality of your paper.

**Research Aim:** The aim of the research is adequately addressed but is rather simplistic, which misses an opportunity for more substantive contributions. I suggest incorporating additional research questions in both the literature review and analysis. For example, would it be feasible to include latent class analysis to examine variations within the population of interest? If possible, can you include the whole sample and other variables to expand your research questions?

**Statistical Analysis:** While the statistical analysis addresses your aims, it is very basic by current standards of scientific rigor and novelty. I recommend including permutation-based methods to complement your research goals, along with non-parametric tests. You also investigate goal-setting behavior and report correlations with data from a previous observational study with a larger sample. However, the measures used in that comparison are not described in the Methods section, and the analysis remains highly descriptive. Also, the analysis needs to be better integrated with the rest of the paper. A more robust approach and statistical modeling, possibly including the entire sample, should be considered.

**Presentation of Results:** The reporting of results should be improved, for instance, statements like “p < 0.00002 (extremely significant)” (line 309) are neither standard nor adequate. Consider using figures to visualize your findings.

**Methods section: ** There is no mention of missing data or how it was handled, nor is it clear if any additional data was collected in your study. Describe all the measures of all variables.

**Study Limitations:** Include a more critical and detailed overview of the limitations of your research design, particularly in assessing only the Big Five personality traits and main demographic variables without controlling for potential confounding variables.

**Suggestions for Further Research:** More substantive suggestions for theoretical development and future research would be beneficial.

Overall, I suggest a major reframing of the whole manuscript.

Only if your next revision can significantly improve on all those suggested elements - and possibly some other aspects that could provide the needed value, it will be considered for publishing.

We look forward to receiving your revised manuscript.

Kind regards,

Srebrenka Letina, Ph.D.

Academic Editor

PLOS ONE

Reviewers' comments:

Reviewer's Responses to Questions

**Comments to the Author**

1. If the authors have adequately addressed your comments raised in a previous round of review and you feel that this manuscript is now acceptable for publication, you may indicate that here to bypass the “Comments to the Author” section, enter your conflict of interest statement in the “Confidential to Editor” section, and submit your "Accept" recommendation.

Reviewer #1: All comments have been addressed

Reviewer #2: (No Response)

2. Is the manuscript technically sound, and do the data support the conclusions?

Reviewer #1: Yes

Reviewer #2: Yes

3. Has the statistical analysis been performed appropriately and rigorously? 

Reviewer #1: Yes

Reviewer #2: Yes

4. Have the authors made all data underlying the findings in their manuscript fully available?

Reviewer #1: Yes

Reviewer #2: Yes

5. Is the manuscript presented in an intelligible fashion and written in standard English?

Reviewer #1: Yes

Reviewer #2: Yes

6. Review Comments to the Author

Reviewer #1: My concerns have been adequately addressed. However, I believe that the process could have been conducted more effectively from the outset, which may have enhanced its scientific value and contributed to a more rigorous outcome.

Reviewer #2: This article presents solid content, is well-conceived, and properly structured and executed.

It offers highly relevant information about the personality traits of multi-marathon runners.

The data collection is extensive, with 593 responses, considering that this is a limited population, such as that of multi-marathon runners.

The statistical analyses are appropriate and very well conducted. They are carried out with an excellent level of detail, including the Bonferroni correction, which is not always performed.

Some results may be unexpected, such as in the dimension of openness to experience. However, for the rest of the results, the expected outcomes are essentially met.

In any case, the work provides an interesting perspective on the psychological characterization of multi-marathon runners. This is of great interest for practical intervention in counseling and psychological coaching for these athletes.

The main challenge of the work, as I understand it, lies in the use of the TIPI instrument to evaluate the Big Five personality dimensions. Although it is a sufficiently standardized and widely accepted tool by the scientific community in personality psychology.

In conclusion, this is a well-conceived and properly executed scientific study, offering relevant contributions to both the theory and practice of sports psychology.

7. PLOS authors have the option to publish the peer review history of their article (what does this mean? ). If published, this will include your full peer review and any attached files.

**Do you want your identity to be public for this peer review?** For information about this choice, including consent withdrawal, please see our Privacy Policy .

Reviewer #1: **Yes: ** Hugo Vieira Pereira

Reviewer #2: **Yes: ** Vanesa García-Peñas

---

## [Author Response · Author response to Decision Letter 2]

7 Jan 2025

Thank you for the opportunity to revise our manuscript and for the detailed comments and suggestions provided by you and the reviewers. We appreciate why these comments were made and we have taken them all on board and have made substantial changes to the manuscript based on this feedback.

In general, we found comments supportive of the submission and appreciate the significant time and effort put in by the editor and reviewers to improve this study. We believe that this study is important not only to academics but also to the professionals and participants that engage with the sport of multi-marathoning. Below we address the concerns point by point and detail the revisions made in the manuscript.

This revised manuscript reflects a significant rework, incorporating substantial additional development and advanced statistical analysis. Specifically, the manuscript has been completely reframed in line with the editors suggestions. Key areas of enhancement include the introduction and integration of a 4th theoretical framework, the Health Belief Model (HMB), increased complexity and depth of the research questions and hypotheses, significant rework on the literary review, introduction of permutation-based methods and other statistical robustness methods, and the integration of Latent Class Analysis which is now pervasive throughout the manuscript. This has meant a significant rework of introduction, methods and results sections and a rewrite of the discussion and conclusion sections. We are confident that these extensive revisions have greatly strengthened the manuscript, addressing all the editor and reviewers comments comprehensively.

A full rebuttal point by point of all the reviewrs comments are included in a comprehensive rebuttal document.

---

## [Decision Letter · Decision Letter 2]

9 Feb 2025

PONE-D-24-41718R2Unveiling the Psychological Traits of Multi-Marathoners: Insights from TIPI Personality Trait AnalysisPLOS ONE

Dear Dr. Lundy,

Thank you for submitting your manuscript to PLOS ONE. After careful consideration, we feel that it has merit but does not fully meet PLOS ONE’s publication criteria as it currently stands. Therefore, we invite you to submit a revised version of the manuscript that addresses the points raised during the review process. A minor revision of the manuscript is required, by addressing the comments made by the new reviewer (reviewer 3, since one of the previous reviewers was not available for this round). In the next revision, in your response to reviewers, explain in detail how you addressed each of the seven points made by reviewer 3.

We look forward to receiving your revised manuscript.

Kind regards,

Srebrenka Letina, Ph.D.

Academic Editor

PLOS ONE

Journal Requirements:

Reviewers' comments:

Reviewer's Responses to Questions

**Comments to the Author**

1. If the authors have adequately addressed your comments raised in a previous round of review and you feel that this manuscript is now acceptable for publication, you may indicate that here to bypass the “Comments to the Author” section, enter your conflict of interest statement in the “Confidential to Editor” section, and submit your "Accept" recommendation.

Reviewer #2: All comments have been addressed

Reviewer #3: All comments have been addressed

2. Is the manuscript technically sound, and do the data support the conclusions?

Reviewer #2: Yes

Reviewer #3: Partly

3. Has the statistical analysis been performed appropriately and rigorously? 

Reviewer #2: Yes

Reviewer #3: Yes

4. Have the authors made all data underlying the findings in their manuscript fully available?

Reviewer #2: Yes

Reviewer #3: Yes

5. Is the manuscript presented in an intelligible fashion and written in standard English?

Reviewer #2: Yes

Reviewer #3: Yes

6. Review Comments to the Author

Reviewer #2: This article presents solid content, is well-conceived, and properly structured and executed.

It offers highly relevant information about the personality traits of multi-marathon runners.

The data collection is extensive, with 593 responses, considering that this is a limited population, such as that of multi-marathon runners.

The statistical analyses are appropriate and very well conducted. They are carried out with an excellent level of detail, including the Bonferroni correction, which is not always performed.

Some results may be unexpected, such as in the dimension of openness to experience. However, for the rest of the results, the expected outcomes are essentially met.

In any case, the work provides an interesting perspective on the psychological characterization of multi-marathon runners. This is of great interest for practical intervention in counseling and psychological coaching for these athletes.

The main challenge of the work, as I understand it, lies in the use of the TIPI instrument to evaluate the Big Five personality dimensions. Although it is a sufficiently standardized and widely accepted tool by the scientific community in personality psychology.

In conclusion, this is a well-conceived and properly executed scientific study, offering relevant contributions to both the theory and practice of sports psychology.

Reviewer #3: Dear Authors,

Your reasoning and course of action is correct, but there are some issues in the reporting of the article that need to be improved, and I will strictly address them in abstraction from the strengths.

1. change the keywords to other than in the title to ensure that your work is more recognizable in the science databases;

2. abbreviate the abstract to relevant issues, the current form is overloaded;

3. the Introduction should be abbreviated by not capturing the cited research, but aptly formulating the content without embellishments;

4. result please improve so that they are only perceptive sentences;

5. explanations of the abbreviations used should be added under the tables;

6. there is a multiplicity of subsections, which should be arranged in Discussion, Limitation, Directions for further research, Practical implications;

7. Conclusions are convictions about the veracity of your results, not a summary. This should be corrected.

8. References should be at most 10 years back, i.e. 2015. This should be corrected. I can suggest known papers on research on personality traits in sports. Consider referring to:

doi: 10.1519/JSC.0000000000002376

doi: 10.1016/j.paid.2020.109973

doi: 10.1002/brb3.2145

doi: 10.3389/fpsyg.2023.1284378

doi: 10.15561/20755279.2024.0302

doi: 10.3389/fpsyg.2024.1428107

7. PLOS authors have the option to publish the peer review history of their article (what does this mean? ). If published, this will include your full peer review and any attached files.

**Do you want your identity to be public for this peer review?** For information about this choice, including consent withdrawal, please see our Privacy Policy .

Reviewer #2: **Yes: ** Vanesa García-Peñas

Reviewer #3: No

---

## [Author Response · Author response to Decision Letter 3]

14 Feb 2025

A very extensive rebuttal document has been submitted labbeled "Response to Reviewers". This specifically takes all of Reviewer 3's comments one-by-one and addresses how these are accomodated within the manuscript.

---

## [Decision Letter · Decision Letter 3]

20 Feb 2025

Unveiling the Psychological Traits of Multi-Marathoners: Insights from TIPI Personality Trait Analysis

PONE-D-24-41718R3

Dear Dr. Lundy,

We’re pleased to inform you that your manuscript has been judged scientifically suitable for publication and will be formally accepted for publication once it meets all outstanding technical requirements.

Kind regards,

Srebrenka Letina, Ph.D.

Academic Editor

PLOS ONE

Additional Editor Comments (optional):

Reviewers' comments:

Reviewer's Responses to Questions

**Comments to the Author**

1. If the authors have adequately addressed your comments raised in a previous round of review and you feel that this manuscript is now acceptable for publication, you may indicate that here to bypass the “Comments to the Author” section, enter your conflict of interest statement in the “Confidential to Editor” section, and submit your "Accept" recommendation.

Reviewer #3: All comments have been addressed

2. Is the manuscript technically sound, and do the data support the conclusions?

Reviewer #3: Yes

3. Has the statistical analysis been performed appropriately and rigorously? 

Reviewer #3: Yes

4. Have the authors made all data underlying the findings in their manuscript fully available?

Reviewer #3: Yes

5. Is the manuscript presented in an intelligible fashion and written in standard English?

Reviewer #3: Yes

6. Review Comments to the Author

Reviewer #3: Dear Authors,

thank you for the corrections made and the answers provided.

Your work has gained significantly in quality.

I recommend the article for publication.

7. PLOS authors have the option to publish the peer review history of their article (what does this mean? ). If published, this will include your full peer review and any attached files.

**Do you want your identity to be public for this peer review?** For information about this choice, including consent withdrawal, please see our Privacy Policy .

Reviewer #3: No

---

## [Editor Report · Acceptance letter]

PONE-D-24-41718R3

PLOS ONE

Dear Dr. Lundy,

I'm pleased to inform you that your manuscript has been deemed suitable for publication in PLOS ONE. Congratulations! Your manuscript is now being handed over to our production team.

Kind regards,

on behalf of

Dr. Srebrenka Letina

Academic Editor

PLOS ONE